Effect of shampoo, conditioner and permanent waving on the molecular structure of human hair

Zhang Yuchen
Alsop Richard J.
Soomro Asfia
Yang Fei-Chi
Rheinstädter Maikel C. rheinstadter@mcmaster.ca
Department of Physics & Astronomy, McMaster University , Hamilton, ON , Canada
Zuo Li
Electronic publication date: 2015 Oct 1
Publication date: 2015
Volume: 3
Electronic Location ID: e1296
Received 2015 Jun 2; Accepted 2015 Sep 15
Copyright: © 2015 Zhang et al.
Copyright year: 2015
Copyright holder: Zhang et al.
License: This is an open access article distributed under the terms of the Creative Commons Attribution License, which permits unrestricted use, distribution, reproduction and adaptation in any medium and for any purpose provided that it is properly attributed. For attribution, the original author(s), title, publication source (PeerJ) and either DOI or URL of the article must be cited.
License URL: https://creativecommons.org/licenses/by/4.0/

Keywords: Hair care products, Human hair, Molecular structure, X-ray diffraction, Shampoo, Conditioner, Permanent waving

Funding: Natural Sciences and Engineering Research Council of Canada National Research Council Canada Canada Foundation for Innovation Ontario Ministry of Economic Development and Innovation NSERC PGS-D Early Researcher Award This research was funded by the Natural Sciences and Engineering Research Council of Canada (NSERC), the National Research Council Canada (NRC), the Canada Foundation for Innovation (CFI) and the Ontario Ministry of Economic Development and Innovation. RJA is the recipient of an NSERC PGS-D. MCR is the recipient of an Early Researcher Award of the Province of Ontario. The funders had no role in study design, data collection and analysis, decision to publish, or preparation of the manuscript.

==============================
The hair is a filamentous biomaterial consisting of the cuticle, the cortex and the medulla, all held together by the cell membrane complex. The cortex mostly consists of helical keratin proteins that spiral together to form coiled-coil dimers, intermediate filaments, micro-fibrils and macro-fibrils. We used X-ray diffraction to study hair structure on the molecular level, at length scales between ∼3–90 Å, in hopes of developing a diagnostic method for diseases affecting hair structure allowing for fast and noninvasive screening. However, such an approach can only be successful if common hair treatments do not affect molecular hair structure. We found that a single use of shampoo and conditioner has no effect on packing of keratin molecules, structure of the intermediate filaments or internal lipid composition of the membrane complex. Permanent waving treatments are known to break and reform disulfide linkages in the hair. Single application of a perming product was found to deeply penetrate the hair and reduce the number of keratin coiled-coils and change the structure of the intermediate filaments. Signals related to the coiled-coil structure of the α-keratin molecules at 5 and 9.5 Å were found to be decreased while a signal associated with the organization of the intermediate filaments at 47 Å was significantly elevated in permed hair. Both these observations are related to breaking of the bonds between two coiled-coil keratin dimers.

Introduction

The human hair has a complex internal structure consisting of many layers including, from outside in, the cuticle, the cortex and the medulla, all bound by the cell membrane complex. These structures and corresponding length scales are shown in Fig. 1. The medulla is a loosely packed, disordered region at the centre of the hair shaft. It is surrounded by the cortex, the structure that makes up majority of the hair fibre and contains keratin proteins and structural lipids. The individual keratin fibres coil together in an organized fashion to form higher level structures, including dimers, protofilaments and intermediate filaments. The cuticle is a layer of overlapping dead cells that form a protective barrier against the outside environment (Robbins, 2012).

Figure 1 Schematics of the X-ray setup and example X-ray data.

The hair strands were oriented in the X-ray diffractometer with their long axis along z. Two-dimensional X-ray data were measured for each specimen covering distances from about 3–90 Å including signals from the coiled-coil α-keratin phase, the intermediate fibrils in the cortex and from lipids in the membrane complex. The 2-dimensional data were integrated and converted into line scans and fit for a quantitative analysis.

Details of hair structure organization have been studied in the past with X-ray diffraction. In particular, the α-helical keratin structure within the cortex has been extensively studied since the 1930s (Astbury & Street, 1932; Astbury & Woods, 1934; Astbury et al., 1935; Pauling & Corey, 1951; Fraser & Macrae, 1958; Fraser, MacRae & Rogers, 1962; Fraser, MacRae & Rogers, 1964; Busson, Briki & Doucet, 1999; Er Rafik, Doucet & Briki, 2004). In addition to X-ray diffraction, other techniques have been employed. For instance, electron microscopy has significantly contributed to studying the organization of intermediate filaments (Fraser et al., 1964). X-ray microdiffraction contributed to exploring the structure of the cuticle, cell membrane complex, lipids and keratin (Busson, Engstrom & Doucet, 1999; Kreplak et al., 1999b; Ohta et al., 2005; Kajiura et al., 2006). Infrared microspectroscopy was used to further elucidate the complex hair structure (Briki et al., 2000).

The development of a breast cancer screening protocol using X-ray diffraction of human hair has been extensively studied by James and Corino (James et al., 2000; Corino & French, 2000; Corino et al., 2009). This screening protocol is already in the clinical trial stage, with an overall accuracy of greater than 77% and a negative predictive value of 99% (Corino et al., 2009). Aside from breast cancer, there exist other diseases that affect hair molecular structure. For instance, Giant Axonal Neuropathy (GAN) is a rare, fatal genetic disease with the presentation of curly hair that significantly differs from that of the parents (Kuhlenbäumer, Timmerman & Bomont, 2014). Similarly, Menkes disease affects hair. Infants having Menkes disease will exhibit short, coarse and twisted hair due to disorders of copper transport (Kaler, 2010). Both these diseases exhibit pili torti, the twisting of the hair shaft by 180°, which is characteristic of defective internal hair structure (Kaler, 2010; Kuhlenbäumer, Timmerman & Bomont, 2014).

There is potential in studying the X-ray diffraction patterns of the hair from individuals afflicted with these diseases in the search of specific disease markers. However, such a diagnostic tool is only valuable, if common hair care products and chemical treatments, such as shampoo, conditioner and permanent waving, do not affect the internal hair structure. Studies in the past have determined that hair care products may affect hair structure, as observed via X-ray diffraction, electron microscopy, and atomic force microscopy (AFM) (Sano, 2006; Nishikawa et al., 1998; Gould & Sneath, 1984; Tate et al., 1993; Bhushan, 2008).

In a previous study, we used X-ray diffraction to analyze the structure of human scalp hair of various individuals with differing characteristics (Yang, Zhang & Rheinstädter, 2014). However, the subjects in this study used differential amounts of shampoo and conditioner on a regular basis. While we found no relationship between this differential use and their respective hair patterns, the comparison was done across various individuals. In this work, we conducted a study on a single subject to solely examine the effect of hair care products and treatments uninfluenced by other factors.

The purpose of this project was to observe the effect that hair products and treatments have on the X-ray profiles using hair from a single subject to maintain baseline consistency in other characteristics. Shampooing, conditioning, perming and various combinations of these treatments were performed on the subject’s hair, the strands were scanned using X-ray diffraction, and the resulting data were analyzed for differences in signals.

Effect of shampoo and conditioner on hair

The main purpose of shampoo is to remove dirt and oil from the surface of the hair fibres and the scalp, while the main purpose of conditioner is to ensure that the hair is smooth for combing. Some commercial shampoos may also have additional components to control dandruff and condition hair. Similarly, conditioners may also prevent static electricity, improve the cosmetic shine and increase protection. Due to the multitude of purposes of these hair care products, they contain a long list of ingredients with various effects on the hair. Shampoos typically contain a primary and a secondary surfactant for thorough cleaning, a viscosity builder, a solvent, conditioning agents, pH adjuster and other non-essential components such as fragrance and colour for commercial appeal. Conditioners usually contain silicone polymers to increase shine and soften hair, cationic polymers such as quaternized nitrogen compounds to reduce static electricity, bridging agents to increase absorption, viscosity builder, pH adjuster and components for commercial appeal (Robbins, 2012; Preedy, 2012).

Since shampoo removes soil and lipids at the surface level, and the active conditioner ingredients only adsorb to the surface, most of the shampoo and conditioning interactions can be expected to occur at the first few layers of the cuticle. The cortex will only be impacted if damage to the cuticle is extensive and the cortex has been exposed. There is evidence that shampooing can result in hair damage in this case; however, the effect is believed to be mostly restricted to the cuticle level. For instance, abrasive action of the shampooing process can damage the keratin and non-keratinous structures at the surface of the hair (Gould & Sneath, 1984). Yet, other studies suggest that the application of conditioner after shampoo increases the protection at the surface of hair fibres, preventing damage from destructive processes such as grooming and bleaching (Tate et al., 1993).

Studies have shown that application of shampoo once can extract approximately 50% of total extractable lipids in hair (Shaw, 1979), and that 70–90% of total lipid can be extracted with repeated shampooing (Koch, 1982). However, it is believed that largely surface level lipids are removed. The cell membrane complex at the cuticle level contains covalently bonded 18-methyl eicosanoic acid (18-MEA) lipids attached to a proteinous cell membrane. At the hair’s outer surface, free lipids within the 18-MEA lipid layers are removed during shampooing (Robbins, 2012). Internal lipids found deeper within the hair shaft are not affected to the same extent as the surface lipids. Internal lipids can travel to the surface layers via the process of diffusion after repeated shampooing (Robbins, 2012).

In summary, shampooing has the effect of extracting surface lipid matter and repeated shampooing could result in damaging structures at the hair surface. These changes could be reflected in the X-ray profiles, particularly the lipid signals. Conditioner ingredients adsorb to hair surface and act as a protective layer preventing this damage. Thus conditioner application, either alone or after the use of shampoo could also affect the X-ray signals.

Effect of permanent waving on hair

Permanent waving is the process of converting straight hair into curled hair using a chemical process. This process takes many steps to complete and involves the breaking and restructuring of the disulfide bonds within hair. In the first step, disulfide bonds are cleaved using reducing agents such as thioglycolates and bisulfites. The reduction permeates throughout the cuticle and into the outer layers of the cortex. Alkaline reductive solutions are used to lift the scales of the cuticle such that the reducing agent can effectively diffuse into the cortex. In a second step, the hair fibres are shaped into the preferred curl and held using curlers. During this time, molecular reorganization occurs in the cuticle layers, as well as in the outer levels of the cortex. Lastly, the fibres are neutralized with an oxidizing agent such as hydrogen peroxide (Robbins, 2012; Bolduc & Shapiro, 2001).

The damaging effects of permanent waving are widely known. Particularly, the use of hydrogen peroxide in the oxidation step increases surface damage and decreases the stiffness of the fibre (Tate et al., 1993). Studies have shown structural changes in both protein composition and lipids after perming (Johnson, 1997). A recent study suggests that prolonged exposure of over 30 min to reducing/oxidizing agents during the permanent waving process reduces the contents of α-helical proteins. This effect is observable in the wide angle X-ray diffraction (WAXD) pattern as a reduction in the intensity of signals related to packing at the level of individual keratin fibres (Nishikawa et al., 1998). The decrease in keratinous protein has also been observed by another group of researchers using a novel method for quantitative analysis of hair components (Kon et al., 1998). Additionally, the permanent waving process was found to extract internal lipids from the cell membrane complex of the fibre, as one study discovered a significant decrease in polar internal lipids after perming treatment (Hilterhaus-Bong, 1989). Moreover, the permanent waving process converts the hydrophobic hair surface to a more hydrophilic one. Thus, in interactions with shampoos, this would result in greater lipid removal from the cuticle (Robbins, 2012).

In summary, permanent waving produces more significant changes to hair structure than shampooing and conditioning, and its effects permeate into the cortical layers of hair fibres. Observable changes to the X-ray profile are, therefore, expected to incur as a result of perming.

Materials & Methods

Preparation of hair samples

This research was approved by the Hamilton Integrated Research Ethics Board (HIREB) under approval number 14-474-T. Written consent was obtained from the participating individual. Samples were obtained from a single subject to eliminate the effect of other potential factors on the observed signals, since the X-ray profiles of various individuals were observed to be different in the previous study (Yang, Zhang & Rheinstädter, 2014). Using samples from a single subject ensures that any changes observed in the signals are the result of the hair treatments. The subject is female, of Asian descent and naturally has slightly wavy and black hair.

The shampoo and conditioner used in this experiment were a common, top-selling brand found in drugstores in US and Canada. When studying the list of ingredients we found that many shampoos contain conditioner ingredients and vice versa, such that a clear distinction is often difficult (Robbins, 2012). The particular brand was chosen for relatively little overlap between shampoo and conditioner ingredients. The active ingredient in the shampoo was sodium laureth sulfate and the active ingredients in the conditioner included cetearyl alcohol, stearamidopropyl dimethylamine and dimethicone. The permanent waving product used was an acid wave product with the main ingredient being glycerolmonothioglycolate (GMT). Since GMT is not stable in water, the reducing formula consisted of two parts that were mixed right before application to produce GMT (Robbins, 2012). The reducing formula was applied to virgin hair, which was subsequently held in position with curling rods. Though the particular product did not require heat application, the hair dresser chose to use an electric heater for thirty minutes to improve treatment result. After heat treatment, the reducing formula was washed off with warm tap water. Then neutralizing lotion was applied to entirety of the head and allowed to settle in for five minutes. Afterwards, the curling rods were removed and hair was rinsed with warm tap water. Lastly, the hair was blow dried.

At the beginning of the study, around 200 hair strands were obtained from the subject’s frontal scalp region, less than 10 cm from the roots. These 200 strands were divided into four approximately equal portions with around 50 strands each. In preliminary experiments where we optimized the X-ray setup and hair preparation we found that a sample that contained ∼50 hair strands gave the best signal. Fewer hairs resulted in faint signals, thicker hair bundles led to absorption and a reduction of signal and increase of background.

Shampoo and conditioner were applied differentially to each of the samples according to Table 1. Then the subject underwent permanent waving treatment in all regions of her scalp hair at a local barber shop using the procedure described above. Another 200 strands were obtained from the same area of the head. These 200 strands were again divided into four 50 strand portions and differential amounts of shampoo and conditioning was applied to each. The treatments applied to each sample can be found in Table 1.

Table 1 Treatment sequences for each of the eight hair samples.

Treatments include cleaning with tap water, shampooing, conditioning and permanent waving. Permanent waving was done directly on the subject’s head in a hair salon while shampooing and conditioning were performed in petri dishes using 10% shampoo and conditioner solutions.

	Perm treatment	Hair care product treatments	
Sample 1	No	Tap water rinse	–	–	–	
Sample 2	Shampoo	Tap water rinse	–	–	
Sample 3	Conditioner	Tap water rinse	–	–	
Sample 4	Shampoo	Tap water rinse	Conditioner	Tap water rinse	
Sample 5	Yes	Tap water rinse	–	–	–	
Sample 6	Shampoo	Tap water rinse	–	–	
Sample 7	Conditioner	Tap water rinse	–	–	
Sample 8	Shampoo	Tap water rinse	Conditioner	Tap water rinse	

Prior to shampoo and conditioning treatments for both non-permed and permed series, the subject did not wash her hair for the duration of one week to facilitate the build-up of lipids on the hair surface. A previous study noted that one application of shampoo can extract 50% of total extractable lipids in hair after not shampooing hair for 2.5–3 days (Shaw, 1979). Thus, the period of a week was determined to be enough time for lipid build-up. Shampoo and conditioning treatments were performed after the hair strands had been collected from the subject’s scalp. Solutions of shampoo and conditioner were prepared in a 1:9 ratio with tap water to give a 10% solution. To shampoo or condition the sample, the hair was held using a tweezers at one end and rinsed in the corresponding 10% solution for 2 min continuously. Then the solution was replaced by tap water and the sample was rinsed for another 2 min. Each sample was allowed to dry for at least 1 hour to eliminate the effect of water on the keratin signals, since water is known to cause swelling of hair (Robbins, 2012). This process was designed to mimic the normal application of shampoo and conditioner during a single shower.

The treated samples were then cut into strands ∼3 cm long. Care was taken to prevent stretching or deforming the hair strands during this process. Each sample was mounted onto a flexible cardboard apparatus as shown in Fig. 2A. The cut-out at the middle of the apparatus allows for the scattering of X-ray signals on the hair sample. The cardboard apparatus is then mounted vertically onto the loading plate of the Biological Large Angle Diffraction Experiment (BLADE) using sticky putty as shown in Fig. 2B. All hair samples were scanned at ambient conditions, i.e., a room temperature and humidity of 28 °C and 50% RH.

Figure 2 Experimental setup.

(A) Apparatus used to mount hair fibres in the experiment; (B) shows how the hair samples were mounted on the X-ray diffractometer.

X-ray diffraction experiment

X-ray diffraction data was obtained using the Biological Large Angle Diffraction Experiment (BLADE) in the Laboratory for Membrane and Protein Dynamics at McMaster University. BLADE uses a 9 kW (45 kV, 200 mA) CuKα Rigaku Smartlab rotating anode at a wavelength of 1.5418 Å. Focusing multi-layer optics provides a high intensity parallel beam with monochromatic X-ray intensities up to 1010 counts/(s × mm2). By aligning the hair strands in the X-ray diffractometer, the molecular structure along the fibre direction and perpendicular to the fibres could be determined. We refer to these components of the total scattering vector, Q→, as qz and q‖, respectively, in the following. An illustration of qz and q‖ orientations is shown in Fig. 1.

The result of such an X-ray experiment is a 2-dimensional intensity map of a large area of the reciprocal space of −2.5 Å−1 < qz < 2.5 Å−1 and −2.5 Å−1 < q‖ < 2.5 Å−1. The corresponding real-space length scales are determined by d = 2π/|Q| and cover length scales from about 2.5 to ∼200 Å, incorporating typical molecular dimensions and distances for primary and secondary protein structures and lipid structures.

Integration of the 2-dimensional data was performed by a script developed in our lab using MATLAB (MathWorks, Natick, Massachusetts, USA). By integrating of the peak intensities along the qz and the q‖ directions, one-dimensional data along each of the two directions were produced. The qz intensity was integrated azimuthally for an angle of 25° over the meridian. The q‖ intensity was integrated azimuthally for an angle of 25°over the equator.

The instrumental background was determined from an empty scan, where no hair was mounted on the apparatus and resulted in a sharp ring at ∼3 Å. This area was, therefore, excluded from the hair data analysis in the following. The corresponding area is covered by a red circle in the 2-dimensional data.

Results

Two-dimensional X-ray intensity maps were collected for all hair samples. These graphs exhibit the characteristic hair signals observed in the previous study (Yang, Zhang & Rheinstädter, 2014). Similar to the previous study, the hair strands were oriented with the long axis of the hair parallel to the vertical z-axis, and the displayed (qz, q‖)-range was designed to cover the length scales of the features of interest. The assignment of biological structures to scattering signals is depicted in Fig. 1. The 2-dimensional data for all hair samples show very good agreement in signal position and shape, suggesting that the hair products used to not have a significant impact on the structure of hair on the observed length scales. However, the perming treatment lead to observable changes in intensity for some signals.

Effect of shampoo and conditioners

Figure 3 show split comparisons between the original untreated hair and hair with various combinations of shampoo and conditioner treatments. The 2-dimensional intensity maps of the respective samples were cut in half and combined such that the left half shows one sample and the right half a different one. This presentation makes a qualitative comparison between the scans easy. The red rings in each of the 2-dimensional plots mark the position of a known background from the apparatus used, as discussed under Materials and Methods. There are no observable differences in the scattering from untreated hair and hair treated with shampoo or conditioner from untreated and shampooed or conditioned hair samples within the resolution of this experiment, suggesting that likely no internal lipid or keratin structures were significantly altered during the application of shampoo and conditioner. A single application of shampoo, conditioner or the combination of the two does not seem to have an effect on the observed signals within the resolution of this experiment.

Figure 3 Split graphs comparing the effects of shampoo and conditioner on internal hair structures.

In (A), the untreated sample is compared with the shampooed sample. In (B), the untreated sample is compared with the conditioned sample. In (C), the untreated sample is compared with the sample with both shampoo and conditioners applied. There were no major changes observed in signals of interest. The red outer ring is the result of a known background from the apparatus and has no relation to hair internal structure.

Effect of perming

The effect of permanent waving on hair structure is illustrated by the split comparison 2-dimensional intensity map in Fig. 4A. Similar to the shampoo and conditioner trials, position of peaks associated with individual keratin and lipids agree between samples. Differences in peak intensities are observed suggesting that perming effects the molecular organization within the hair.

Figure 4 Split graphs comparing the effects of shampoo and conditioner on permanently waved hair.

In (A), the untreated sample is compared with the permed hair. In (B), the permed hair is compared with the permed and shampooed sample. In (C), the permed hair is compared with the permed hair treated with conditioner and in (D) the permed sample is compared with the permed sample with both shampoo and conditioners applied. Similar to Fig. 3, the red outer ring is the result of the instrumental background.

There are two observable differences between the untreated and the permed samples at the small angle q‖ region, as shown in Fig. 5. The signals corresponding to this region are best visualized using the small angle X-ray diffraction (SAXD) technique to quantitatively analyze the position and amplitude of signals. The signals at 90 Å, 47 Å and 27 Å were observed in the q‖ direction of the SAXD data, in agreement with the results of the previous study (Yang, Zhang & Rheinstädter, 2014). As seen in Fig. 5, the difference between untreated and permed hair observed on the 2-dimensional maps and the line graphs is represented by the disproportional increase in the 47 Å signal and slight reduction of the 27 Å signal after perming. This is expected to be the region were keratin intermediate filaments demonstrate packing along the cross-sectional plane of the hair into macro-fibrils (Yang, Zhang & Rheinstädter, 2014). This suggests that structural changes do occur at the intermediate filament level after permanent waving.

Figure 5 Small angle X-ray scattering (SAXS).

(A) Two-dimensional intensity map of untreated hair vs permed hair. While the majority of signals corresponding to keratin fibres and lipids remain unchanged, the area highlighted by the close-up view of (A) showed slight differences at the region corresponding to intermediate filaments. (B) and (C) the SAXD data of untreated hair vs. permed hair. Peaks are shown at 90 Å, 47 Å and 27 Å. Increased 47 Å peak and decreased 27 Å peak intensities are observed.

Another change is observed after integrating the WAXD data along both the q‖ and qz directions. In the q‖-direction, the peak at 9.5 Å is noticeably decreased in comparison to the 4.3 Å peak after perming. In the qz-direction, the peak at 5.0 Å is also decreased in comparison to the peak at 4.3 Å. This change is visualized in Fig. 6. Both of these peaks refer to the coiled-coil protein structure, i.e., the α-helical keratin packing distances, while the peak at 4.3 Å refers to lipid packing. The peak at 2.08 Å−1 or ∼3 Å is the result of instrumental background and has no relation to hair structure.

Figure 6 Wide angle X-ray scattering (WAXS).

(A) The wide angle X-ray diffraction data for the qz direction. There is a noticeable decrease of the 5.0 Å signal corresponding to α-keratin helix distances in comparison with the lipid signal at 4.3 Å. (B) The wide angle X-ray diffraction data for the q‖ direction. There is a noticeable decrease of the 9.8 Å signal corresponding to α-keratin helix distances in comparison with the lipid signal at 4.3 Å.

Shampoo and conditioner usage is often advertised to repair damage to hair (“liquid hair repair,” “advanced care and repair” or “keratin repair shampoo”). In this study, we applied combinations of shampoo and conditioner to permanently waved hair to examine whether these hair care products had an impact on the changes incurred during perament waving. As shown in Figs. 4B–4D, the signal in the SAXD region at 47 Å remained after a single application of shampoo or conditioner. Thus, a single application of shampoo and conditioner was found to have no restorative effect to the changes sustained by hair fibres during permanent waving. Moreover, the wide angle region of the 2-dimensional data remains unchanged. As with untreated hair, a single application of shampoo or conditioner to permed hair did not result in changes to internal hair structure on the length scales of the intermediate filaments or keratin proteins studied in this experiment.

Discussion

Shampoo and conditioner appear to have no observable effect on the internal keratin and lipid signals visualized using X-ray diffraction. This is in congruence with the fact that these hair care products should solely act at the outer cuticle layer of the fibre. The potential extraction of free lipids by a single use of shampoo is not significant enough to be observed. It is also possible that the lipid signals observed in the experiment scatter from covalently bonded internal lipids rather than free lipids. Moreover, the single use of shampoo is not expected to damage cortical structures. Damage to hair reported from shampoo likely arises from repeated applications, leading to erosion of surface structures. The data also suggests that the adsorption of conditioning ingredients to the outer surface of hair fibres will not impact the X-ray signals.

Since the visualized X-ray profiles are not affected by application of shampoo and conditioner, it is possible to use this method to study the disorganization of keratin fibres without concern over subjects’ differential use of these hair care products.

However, for permanently waved hair samples, differences were observed in the intensities of signals. At the wide angle region, there is a reduction of α-helical signals seen in the treated samples along both equatorial and meridional directions in comparison to the lipid signals. The keratin proteins in the cortex are known to organize in bundles, whose structure is dominated by α-helical coiled-coils (Pauling & Corey, 1950; Pinto et al., 2014; Yang et al., 2014; Yang, Zhang & Rheinstädter, 2014). Coiled coils consist of α-helices wound together to form a ropelike structure stabilized by hydrophobic interactions. The coiled-coil motif is found in about 10% of the proteins in the human genome (Neukirch, Goriely & Hausrath, 2008).

The main features of this motif is a ∼9.5 Å (corresponding to q‖ ∼ 0.6 Å−1) equatorial reflection corresponding to the spacing between adjacent coiled-coils and a ∼5.0 Å meridional reflection (corresponding to qz ∼ 1.25 Å−1) corresponding to the superhelical structure of α-helices twisting around each other within coiled-coils (Crick, 1952; Cohen & Parry, 1994; Lupas & Gruber, 2005). These signals were observed in the X-ray data. A decrease of the corresponding peak intensities in permed hair is not related to a decrease in the number of keratin proteins in the hair, however, indicative of a reduced number of coiled-coil proteins, i.e., indicative of a breaking of the bonds between two coiled dimers. We note that any X-ray diffraction experiment is not sensitive to protein monomers but the scattered signals stem from protein dimers. Perming products, therefore, seem to deeply penetrate the hair and break up hydrophobic bonds between coiled-coil keratin proteins. The basic building block of each intermediate filament is such a dimer of a coiled-coil pair of proteins.

The SAXD signals observed in this experiment agree with those of previous experiments. Signals in the SAXD region of 90 Å, 47 Å and 27 Å along the cross-sectional direction have long been recognized as representative of distances between microfibrils immersed in a sulfer rich matrix (Er Rafik, Doucet & Briki, 2004). Many models have been developed in the past 50 years with an attempt to characterize the arrangement of microfibrils in the formation of intermediate filaments, with some assuming a ring-core arrangement while others stating a uniform profile (Er Rafik, Doucet & Briki, 2004). Yet the actual structure of intermediate filaments remain elusive and different model are used to describe SAXD profiles in the literature.

The signal at 47 Å was found to be significantly increased in the permed samples in Fig. 5. This effect was observed in all four permed samples in comparison to the non-permed samples. This is in agreement with the observation that the permanent waving process produces changes at the intermediate filament level (Robbins, 2012).

The small angle peak pattern in Fig. 5 with peaks at 90 Å, 47 Å and 27 Å is compatible with a 7 dimers arranged on a hexagonal lattice, as shown in Fig. 7A. The structure factor of this structure was calculated and shows excellent agreement with the experimental data. Perming was found to have two effects on the hair structure: (1) a decrease of the coiled-coil signal indicative of a reduced number of protein dimers and (2) an increase of the 47 Å signal. The fact that the small angle peak positions do not change after perming is indicative that the overall structure of the intermediate filaments is not affected by perming. However, decomposition of the keratin coiled-coils may lead to the internal fibre structure shown in Fig. 7B, where individual keratin molecules arrange in a hexagonal pattern.

We have calculated the diffraction patterns of different filament arrangements. The small angle signals in our experiments are consistent with a hexagonally packed organization of dimers, as shown in Fig. 7A, with a spacing between the dimers of 90 Å. Decomposition of the keratin coiled-coils into single helices may change the internal fibre structure as shown in Fig. 7B, where individual keratin molecules arrange in a hexagonal packing. The corresponding calculated peak pattern results in an increase in the 47 Å signal, in agreement with the experimental findings. While the calculations assumed that all dimers split into monomers, the slight decrease in the coiled-coil peaks indicate that most proteins still form coiled-coils.

Technically, packing of the filaments in non-permed hair was described by a primitive monoclinic unit cell, space group P2, with lattice constants a = c = 90 Å and α = 90°, β = 60° and γ = 90°. After perming, a base centered monoclinic lattice, space group C2, with lattice constants a = c = 90 Å and α = 90°, β = 60° and γ = 90° well describes the experimental findings. The corresponding diffraction patterns were calculated and convoluted with the instrumental resolution, which leads to a finite peak width. The results are shown in Fig. 7 and the corresponding [hkl] reflections are marked in the figure. The pronounced change between the two peak patterns is the increase in the [200] reflection, indicative of the occurrence of monomers at positions between two dimers.

Figure 7 Calculated diffraction patterns.

(A) A hexagonal packing of the coiled-coils well describes the experimental small angle data in Fig. 5B in non-treated hair. (B) The second peak was found to be increased in intensity after perming, consistent with a structure where some coiled-coils are dissolved into monomers and occupy positions in between the dimer locations. [hkl] indices of the primitive and base centred monoclinic lattices used for the calculation are marked. Cartoons show the arrangements of dimers and monomers within the filaments.

In future studies involving subjects with recent permanent waves, care must be taken to rule out the smaller intensity of the coiled-coiled keratin signals and the increased intensity of the 47 Å signal as a possible structural defect. However, other abnormalities observed in the profile, such as a split, extra or shifted signal could be assigned to structural abnormalities independent of the perming treatment.

Conclusions

X-ray diffraction is able to visualize signals corresponding to keratin coiled-coils, intermediate filaments and lipids in the cell membrane complex. The hair structure of a single individual under shampoo, conditioning and permanent waving treatments was studied in the natural state using wide angle and small angle X-ray diffraction. Signals corresponding to the coiled-coil structure of keratin fibres and lipids from the cell membrane complex identified previously study were observed. For shampooing and conditioning treatments, no differences were observed before and after treatment within the resolution of this experiment, suggesting that their effects are limited to the surface of the hair fibre, leaving the inner cortex unchanged. However, permanent waving is a more invasive chemical treatment that results in both a decrease in α-keratin signals and the changes of a signals at the intermediate filament level related to breaking of the bonds between two coiled dimers. Otherwise, the general shape of both equatorial and meridional profiles remain unchanged. Thus, use of hair care products, such as shampoo and conditioners, would not impact the molecular structure of hair, while permanent waving treatments lead to slight alterations. These are aspects that one should keep in mind when interpreting the results of X-ray diffraction profiles in relation to structural abnormalities.

Supplemental Information

Supplemental Information 1 Raw data

Click here for additional data file.

Additional Information and Declarations

Competing Interests

Author Contributions

Human Ethics

The authors declare there are no competing interests.

Yuchen Zhang and Richard J. Alsop conceived and designed the experiments, performed the experiments, analyzed the data, contributed reagents/materials/analysis tools, wrote the paper, prepared figures and/or tables, reviewed drafts of the paper.

Asfia Soomro performed the experiments, contributed reagents/materials/analysis tools.

Fei-Chi Yang contributed reagents/materials/analysis tools, prepared figures and/or tables.

Maikel C. Rheinstädter conceived and designed the experiments, performed the experiments, wrote the paper, prepared figures and/or tables, reviewed drafts of the paper.

The following information was supplied relating to ethical approvals (i.e., approving body and any reference numbers):

Hamilton Integrated Research Ethics Board (HIREB) under approval number 14-474-T.

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
