# Peer review of "Effect of shampoo, conditioner and permanent waving on the molecular structure of human hair"

_PeerJ, doi:10.7717/peerj.1296_

## Round 0.1 · original submission · Major Revisions

Please carefully address the reviewers' concerns. Thank you.

Reviewer 1 ·

Basic reporting

1. Please state the purpose or application of this study. Is it a useful prediction of disease such as breast cancer (mentioned in the introduction part)? Please expand a bit more on the significance of this study. Otherwise, it seems that the topic is not very interesting.

2. It is not scientific to state that 'Genetics was found to play an important role since nearly identical patterns were observed for identical twins and the father and daughter pair, but not for the fraternal twins [17]' It is not necessary that fraternal twins are more genetically different than the father and daughter without showing any evidence of their genetic background. Please be aware of the expression even though cited from an already published paper.

Experimental design

The description of experiment design was vague. What is the difference between this study and the previous study in ref 17? As mentioned in the introduction part:' Our previous study found no direct evidence for an effect of hair care products on observed signals [17]. Yet, the effect of hair care products on a single individual's hair has yet to be observed', the review assume that the previous study take the average data from multiple samples of different individuals. However, and in the method part, the author stated that ' Written consent was obtained from all participating individuals. Samples were obtained from a single subject .... .' This is quite confusing. Please explain.

Validity of the findings

No Comments.

Additional comments

In this study, the authors were trying to study the effect of shampoo, conditioner, and permanent waving on the molecular structure of human hair. However, there are several issues required to be addressed before consideration of publication.
1. Please state the purpose or application of this study. Is it a useful prediction of disease such as breast cancer (mentioned in the introduction part)? Please expand a bit more on the significance of this study. Otherwise, it seems that the topic is not very interesting.
2. It is not scientific to state that 'Genetics was found to play an important role since nearly identical patterns were observed for identical twins and the father and daughter pair, but not for the fraternal twins [17]' It is not necessary that fraternal twins are more genetically different than the father and daughter without showing any evidence of their genetic background. Please be aware of the expression even though cited from an already published paper.
3. The description of experiment design was vague. What is the difference between this study and the previous study in ref 17? As mentioned in the introduction part:' Our previous study found no direct evidence for an effect of hair care products on observed signals [17]. Yet, the effect of hair care products on a single individual's hair has yet to be observed', the review assume that the previous study take the average data from multiple samples of different individuals. However, and in the method part, the author stated that ' Written consent was obtained from all participating individuals. Samples were obtained from a single subject .... .' This is quite confusing. Please explain.

Reviewer 2 ·

Basic reporting

Overall, it is a well-written paper with sophisticated research. The study topic is quite interesting and relative to us. The purpose of this study is to exam the effect of hair products and treatment on the molecular hair structure. However, lack of novelty is the main concern of this paper although authors implied that using single subject to collect samples to maintain baseline consistency is a unique approach. However, it does not provide a strong stand to support he novelty nature of this study.

Results: The description of the figures mentioned at the results section is redundant since it is supposed to be included in the caption section.

Discussion: There is a large proportion of the reference from year 1950-1999. More updated research should be included in this paper to improve the relevance and quality of the study. Line 478-485. Author mentioned following permanent waving the signal at 47 Å was increased which is in conflict with the speculation based on previous studies while signal at 27 Å was decreased as expected. However lack of discussion in depth to explain this controversy.

Experimental design

No Comments

Validity of the findings

Could you please provide the rationality of using around 200 hair strands for each group instead of 300 or 400?

Additional comments

Overall, it is a well-written paper with sophisticated research. The study topic is quite interesting and relative to us. The purpose of this study is to exam the effect of hair products and treatment on the molecular hair structure. However, lack of novelty is the main concern of this paper although authors implied that using single subject to collect samples to maintain baseline consistency is a unique approach. However, it does not provide a strong stand to support he novelty nature of this study.

Results: The description of the figures mentioned at the results section is redundant since it is supposed to be included in the caption section.

Discussion: There is a large proportion of the reference from year 1950-1999. More updated research should be included in this paper to improve the relevance and quality of the study. Line 478-485. Author mentioned following permanent waving the signal at 47 Å was increased which is in conflict with the speculation based on previous studies while signal at 27 Å was decreased as expected. However lack of discussion in depth to explain this controversy.

---

## Round 0.2 · Minor Revisions

Please address the reviewer's comment regarding the protocol change. Thanks.

Reviewer 2 ·

Basic reporting

In general, this manuscript was significantly improved after this revision. Particularly, the added section (related to the significance and application of this study in the introduction section), figure 7, and the in-depth explanation of the discrepancy of the results did strengthen the consideration for publication.

However, the "200 strands" collected from a single subject did NOT comply to what you stated in the consent form (see below).
"Why am I being invited to donate this sample?
You are being invited to donate some hair strands (approximately 10 strands) for the analysis of the molecular structure of your hair. Your information will be used to compare the structure of human hair
between different individuals and hair of different visual appearance, such as colour or waviness."
Any changes and amendments to the protocol must be approved by HIREB, please provide the evidence of the second approval.

Experimental design

No Comments

Validity of the findings

No Comments

Additional comments

No Comments

---

## Round 0.3 · accepted · Accept

I am glad to inform you that your paper has been accepted.